# Folic acid retention evaluation in preparations with wheat flour and corn submitted to different cooking methods by HPLC/DAD

**Emmanuela Prado de Paiva Azevedo**[1☯‡*], **Eryka Maria dos Santos Alves**[2☯], **Samuel de Santana Khan**[3☯], **Leonardo dos Santos Silva**[4☯], **José Roberto Botelho de Souza**[5☯], **Beate Saegesser Santos**[6☯], **Carlos Bôa-Viagem Rabelo**[7☯], **Ana Carolina dos Santos Costa**[1☯], **Clayton Anderson de Azevedo Filho**[8☯], **Margarida Angélica da Silva Vasconcelos**[2☯‡]

**1** Rural Technology Department, Federal Rural University of Pernambuco, Recife, PE, Brazil, **2** Nutrition Department, Federal University of Pernambuco, Recife, PE, Brazil, **3** Consumer Sciences Department, Federal Rural University of Pernambuco, Recife, PE, Brazil, **4** Biology Institute, University of Pernambuco, Recife, PE, Brazil, **5** Zoology Department, Federal University of Pernambuco, Recife, PE, Brazil, **6** Pharmacy Department, Federal University of Pernambuco, Recife, PE, Brazil, **7** Zootechnics Department, Federal Rural University of Pernambuco, Recife, PE, Brazil, **8** ASCES College—Caruaruense Association of Higher Education and Technical (Mantenedora), University District, Caruaru, PE, Brazil

☯ These authors contributed equally to this work.
‡ EPPA and MASV are the Senior Authors.
* paiva.ufrpe@gmail.com

## Abstract

Folic acid content was evaluated in food preparations containing wheat and corn flour submitted to baking, deep-frying, and steaming. Commercially fortified flours showed the absence of folic acid. Flours with laboratory folic acid fortification showed 487 and 474 μg of folic acid in 100 g of wheat and corn flours, respectively. In the corn flour preparations, the cake had the highest retention (99%) when compared to *couscous* (97%). Besides, the cake showed higher retention when compared to the wheat flour preparations due to the interactions of the folic acid with the hydrophobic amino acids of the Zein, a protein found in corn. In wheat flour preparations, vitamin retention was 87%, 80% and 57% in bread, cake, and White sauce respectively. These findings relate to the change of the physicochemical properties of food components that occurs during mixing and cooking of the ingredients.

## 1. Introduction

Folic acid is a yellow crystalline compound synthetically produced by the food industry which is used in nutritional supplements and fortification of cereals and flours [1, 2]. Also found in food, vitamin B9 plays an important role in the synthesis of proteins and haemoglobin, cell growth, hypertension control, and prenatal development. Its deficiency has been pointed out as a factor that induces the appearance of serious diseases that affect human beings such as congenital malformations, cardiovascular diseases, cerebrovascular diseases, among others [3, 4].

**Data Availability Statement:** All relevant data are within the paper and its Supporting Information files.

**Funding:** The author(s) received no specific funding for this work.

**Competing interests:** The authors have declared that no competing interests exist.

The high amount of problems related to folic acid deficiency led several countries to initiate the fortification of cereals and flours voluntarily or through legislation [1, 5]. In Brazil, the requirements for the fortification of wheat and corn flour were updated in April 2017 through the publication of RDC (Collegiate Board Resolution) No. 150 of April 13, 2017 [6], which revoked Ordinance SVS/MS (Department of Health Surveillance–Ministry of Health) n. 344/2002 [7] and established, among other updates, a maximum limit (220 μg.100g$^{-1}$) for the addition of the vitamin in order to avoid exposure of the population to unnecessary risk [6].

The fortification of commonly consumed food items, such as wheat flour and corn flour, has been proposed as the best method to ensure the increase in folic acid intake in order to reduce the risks associated with micronutrient deficiency [8, 9]. However, after food enrichment, nutrient behavior studies are necessary to evaluate the present vitamin content after processing, storage and cooking of the fortified food since such conditions may alter the concentration of the nutrients [10]. As to cooking, the thermolabile nature of the vitamins contributes to the fact that the heat treatment is an important factor for the degradation of these nutrients, and its effect may vary according to the type of food, the chosen method and cooking time [11].

Folic acid is an unstable molecule, it is light sensitive and undergoes oxidative degradation, which can be reinforced by pH, oxygen, heat and acidic conditions [3], but little is known about its stability when facing different cooking methods.

Therefore, the aim of the present study was to investigate the folic acid content and the folic acid retention in food preparations made with corn flour and wheat flour, submitted to cooking methods usually applied in a household.

## 2. Material and methods

### 2.1. Samples and reagents

Samples of flours and ingredients used to prepare the food preparations were acquired from supermarkets (Assaí wholesaler, Hiper Bompreço and Extra supermarkets) located in the city of Recife–PE, Brazil. All ingredients used in the preparation, including flour, are produced and marketed in Northeastern Brazil and comply with current legislation. The brands available for analysis were, wheat flour: Dona Bente, Rosa Branca, Finna, Boa Sorte and Sarandi and Cornflour: Coringa, Novomilho, Santa Clara and Vitamilho. The sample quota related to commercially fortified flours was composed of six different brands of wheat flour (WF) and three different brands of corn flour (CF) that were collected from two different batches and evaluated in different months of the year (July and September), totalizing the value of 18 samples. The laboratory folic acid fortified wheat flour and corn flour had the addition of the vitamin following the fortification protocol described in item 2. and they were also used in the elaboration of the different preparations for the study of folic acid retention.

Methanol was purchased from Merck®, Brazil; monobasic potassium phosphate by Sigma-Aldrich®, Germany; orthophosphoric acid by Fluka®, Switzerland and ammonium acetate and ascorbic acid by Vetec®, Brazil, all of them in analytical grade (greater than 99%). The standard for folic acid identification and quantification was obtained from Sigma-Aldrich®, U.S.A. The water used was obtained from a Milli-Q ultra-purification water system (Millipore, Bedford, MA, U.S.A).

### 2.2. Protocol for laboratory folic acid fortification of wheat and corn flour

Approximately 5,000 μg (0.0050 g) of the folic acid standard were weighed and added to 500 g of wheat flour or 500 g of corn meal in order to achieve a concentration of 1000 μg of folic acid for every 100 g of wheat or corn flour. For a good homogenization of the vitamin in the

farinaceous matrix, two procedures of homogenization were used: (1) Folic acid (0.0050 g) was added to the total content (500 g) of the farinaceous matrices, shaken in a cutter (Metvisa) for 8 minutes and sieved using a domestic sieve; (2) The 500 g of the wheat and corn flour to be fortified were first shaken in a cutter (Metvisa) for 1 minute and then sieved, discarding the unsifted residue. This procedure was performed three times. After that, the total content of the flours (500 g) was fractionated into five 100 g portions. The total amount of folic acid (0.0050 g) to be added to the farinaceous matrices was also weighed and divided into fractions (5 fractions of 0.0010 g). In another essay,Each folic acid fraction (0.0010 g) was added to each portion (100 g) of the wheat and corn flours and the mixtures (flour and vitamin) were first homogenized manually for 1 min using a spatula, and then homogenized in a cutter (Metvisa) also for 1 minute. This procedure was repeated for each of the five portions (100 g each) from the fractionation of the total content (500 g) of the wheat and corn flours that would be fortified. After each portion of the flour was added folic acid and homogenized, manually and mechanically, the total content (500 g) was mixed and homogenized in cutter for 2 minutes. At the end of the fortification procedure, the total time used for the homogenization of the vitamin into the farinaceous matrix, including manual (1 min for each 100 g) and mechanical homogenization (1 min for every 100 g + 2 minutes for 500 g), was of 12 minutes.

The fortified flours were then packed under the same conditions as those applied in industry and were used afterward to determine folic acid content as well as to prepare the different food preparations used to evaluate folic acid retention.

## 2.3. Food preparations

The preparations were prepared with laboratory folic acid fortified wheat flour and corn flour, from pre-tested recipes, following the preparation techniques described below:

## 2.4. Wheat flour cake

The egg whites of two large eggs (50 g each) were separated from the yolks and beat until firm peaks were obtained. The egg yolks from the two eggs, 40 g of margarine and 160 g of sugar were mixed with a domestic food mixer until a homogeneous mixture was obtained. 240 mL of whole milk and 240 g of wheat flour were gradually added to the egg yolk, margarine and sugar mixture as it was still being mixed. Finally, 30 g of baking powder and the whipped egg white were added and mixed. The cake mixture was poured into a greased and floured Bundt pan. The conventional oven was used for baking. It was preheated and the cake was baked at 180˚C for 40 minutes.

## 2.5. White cream sauce

One medium garlic (5 g) and ½ of one small onion (25 g) were grated and fried in 10 g of margarine. The 240 mL of whole milk, 20 g of wheat flour and salt to taste were added gradually and stirred until a thick cream was obtained. The heat was lowered from about 100˚C to 60˚C and the white sauce was left to cook for five minutes, stirring when necessary. The pan was removed from the heat, followed by the addition of 50 g of table cream to the sauce. The white cream sauce was cooked in the pan, in a conventional stove at 60˚C for 5 minutes.

## 2.6. Bread Loaf

In a small bowl, 10 g of yeast was mixed with 1/4 cup of warm water until they were dissolved. The mixture was left to rest for about 5 minutes until bubbles started to rise. In a large bowl, 240 g of the laboratory fortified wheat flour and 8g of salt was mixed leaving a well in the center

of the mixture. The dissolved yeast was gradually added to the mixture of flour and salt, from the center outwards. The dough was brought together using the hands and 300 mL of milk was also added as the ingredients were being mixed. As soon as the flour absorbed the liquids, the procedure was followed by the incorporation of butter to the dough and more mixing. The dough was pressed, kneaded, stretched and kneaded again for 10 minutes, until it obtained a soft and moist texture. It was then shaped in the shape of a ball and returned to the bowl, covering it with cling film, and left to rest for 1 hour until it doubled in size. After that, the dough was transferred to the workbench and divided in half, flattening each half into a rectangle. To make the bread loaves, one of the two larger sides of the rectangles was folded to the center of the dough and tightened gently; it was then covered by its opposite side and tightened well to seal. The two remaining sides of the rectangle (the smaller ones) were then folded 4 cm from the end of the dough and the seams were pressed tightly.

The dough was transferred, with the dough seam side down, to a butter greased baking pan. It was covered with cling film and set to rest and prove for another 40 minutes. Exactly 20 minutes before the proofing time was finished, the oven was preheated to 180°C. The dough was baked in a conventional oven for about 40 minutes.

### 2.7. Corn cake

In a domestic blender, 75 mL of whole milk, 85 mL of corn oil and 2 large eggs were put together and blended well. After that, the blender was turned off, 80 g of white sugar was added and the mixture was blended a bit more. Following the sugar, the blender was turned off once more for the addition of 240 g of fortified corn flour and the mixture was blended once again. Salt to taste and 5 g of baking powder were added to the mixture and it was blended again. The mixture was poured into a greased and floured Bundt pan. The cake was baked in a preheated conventional oven at a temperature of 180°C for 40 minutes.

### 2.8. Couscous

120 g of folic acid laboratory fortified corn flour and 2 g of salt were placed in a glass bowl. 200 mL of water was gradually added and mixed well with the aid of a spatula. After mixing everything, the couscous mixture was placed in a couscous steamer and taken to the heat. The couscous was cooked in a conventional stove at an initial temperature of 100°C, being reduced to 60°C when the couscous steamer water began to boil. At this temperature, it was allowed to steam for 9 minutes before the heat source was turned off.

## 3. Folic acid analysis by HPLC

### 3.1. Sample extraction

The folic acid was extracted from the samples (fortified wheat flour and corn flour, raw preparations, and cooked preparations) following the methods of Paiva et al. [12]. Approximately 2 grams of the sample was weighed and put into a falcon test tube and 20 mL of the ammonium acetate extracting solution (50 mmol $L^{-1}$) was added, which were then vortexed for one minute. The extracts were submitted to a thermostated water bath treatment at 40°C for 10 minutes. Then the samples were centrifuged in at 5000 g rotation at a temperature of 4°C for 15 minutes with subsequent filtration on cellulose acetate membranes of 0.22 μm porosity. The amount equivalent to 1 mL of each extract was filtered once more into polyvinylidene fluoride membranes of 0.22 μm porosity into vials which were then brought to the chromatograph. All determinations were performed in quintuplet.

## 3.2. Chromatographic conditions

All High-Performance Liquid Chromatography (HPLC) analyses were performed on a Shimadzu liquid chromatography (Shimadzu®, Japan), consisting of the following modules: controller system, model CBM-20A; diode array detector, model SPD-20AV; quaternary pump, model LC-20TA; column oven, model CTO-20AC; autosampler, model SIL-20AC. All commands were performed using LC Solution software (version 1.25, Shimadzu®, Japan).

The mobile phase, used in isocratic mode, was composed of a mixture of monobasic potassium phosphate buffer (pH 2.0) and 85% methanol (85:15 v/v). The flow rate was 0.5 mL/min, with an injection volume of 20 μL and a total run time of 20 min. Folic acid was eluted with a retention time of approximately 18.2 minutes. The column was washed with water and conditioned with acetonitrile (Merck-grade HPLC) at the end of the analyses (PAIVA et al., 2012).

## 3.3. Identification and quantification

Peak identification was performed by spectral similarity (an equipment feature that allows the comparison between the standards' spectra and samples' spectra) and by comparing the retention time in both standards and the samples. Quantification was performed using external standardization by constructing an analytical curve with 10 different concentrations of folic acid ranging from 0.05 μg.mL$^{-1}$ to 100 μg.mL$^{-1}$. The solutions used to draw the calibration curve were individually prepared by diluting the stock solution (1000 μg.mL$^{-1}$ folic acid) in ammonium acetate (8 mmol.L$^{-1}$) from the individual standard.

The quantification limit (QL) was defined as the lowest concentration of the analyte that reached a signal 10 times above the baseline noise [13], using the following equation: *QL = (SD x 10) / SC*. The detection limit (DL) was defined as the lowest concentration of the analyte that reached a signal 3 times above the baseline noise [13], using the following equation: *DL = (SD x 3) / SC*. SD is the standard deviation of the intercept with the y-axis of at least three calibration curves and SC is the slope of the calibration curve.

## 3.4. True retention

The retention of folic acid was calculated by the percentage of vitamin retention in the food preparations made with the laboratory fortified wheat flour and corn flour, using different cooking methods. The percentage was obtained considering the weight changes after the raw mixture was cooked. For this, the true or real retention formula (% RR) [14] was used as described below:

$$\% \, RR = \frac{(\text{mg of folic acid per g of cooked preparation x weight of cooked preparation})}{(\text{mg of folic acid per g of raw preparation x weight of raw preparation})} \, x \, 100$$

## 3.5. Statistical test

Initially, normality and homoscedasticity tests were carried out (Bartlet's test). In the absence of such parameters for a few samples, the information was transformed into a log$^{(x+1)}$ to meet the requirements of analysis of variance (ANOVA) to observe the differences between treatments. For the independent sample treatments classified as dry-heat and moist-heat, the Kruskal-Wallis test was used. *Statistica 7.0* software was used for all statistical analyses.

# 4. Results and discussion

As can be seen in Fig 1, in samples analysed in July and September, there was no signal in time and absorptivity matching the folic acid standard, making us conclude that the samples obtained from the local markets did not have the vitamin. It is important to emphasize that

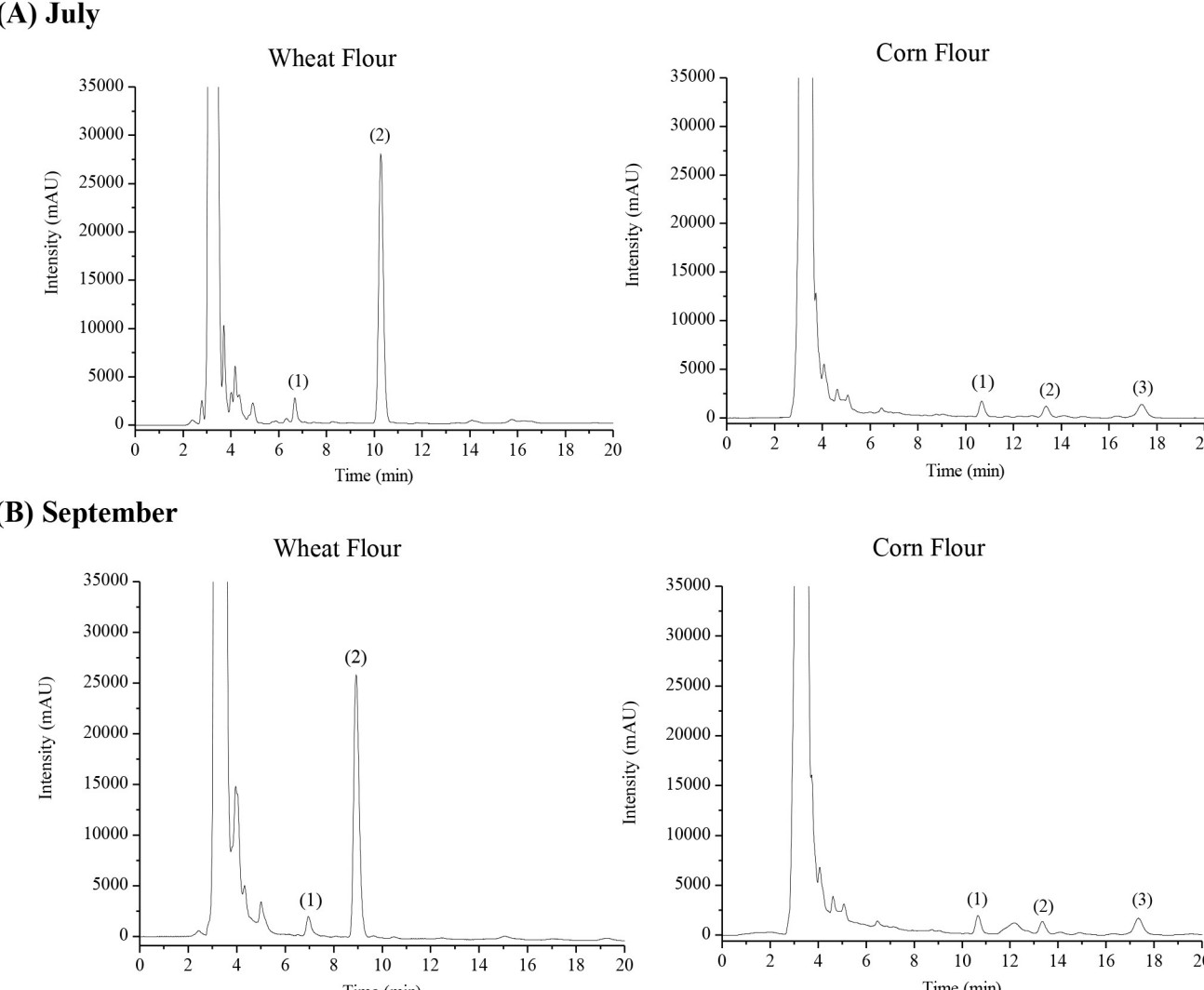

**Fig 1.** HPLC chromatogram of commercially fortified wheat flour and corn flour evaluated in July (A) and September (B). Chromatograms for one of the samples of commercially fortified wheat flour and corn flour evaluated in July (A) and one of the wheat flour and corn flour samples evaluated in September (B).

other (unpublished) studies were done in previous years and folic acid was not found in samples from the Northeast of Brazil. The chromatograms shown have signals, other than folic acid, that couldn't be identified in this study. Yet, they have a high similarity with the folates database available in our software.

## Determination of folic acid in commercially fortified and laboratory fortified wheat and corn flours

In Fig 2 the chromatograms of the laboratory fortified wheat and corn flours are illustrated in the chromatograms and in Table 1 the content of laboratory fortified folic acid in the corn and wheat flours are shown.

The laboratory fortified folic acid flour samples had a medium concentration of 487 µg.100g$^{-1}$ for the wheat flour and 474 µg.100g$^{-1}$ for the corn flour as seen in Table 1. The

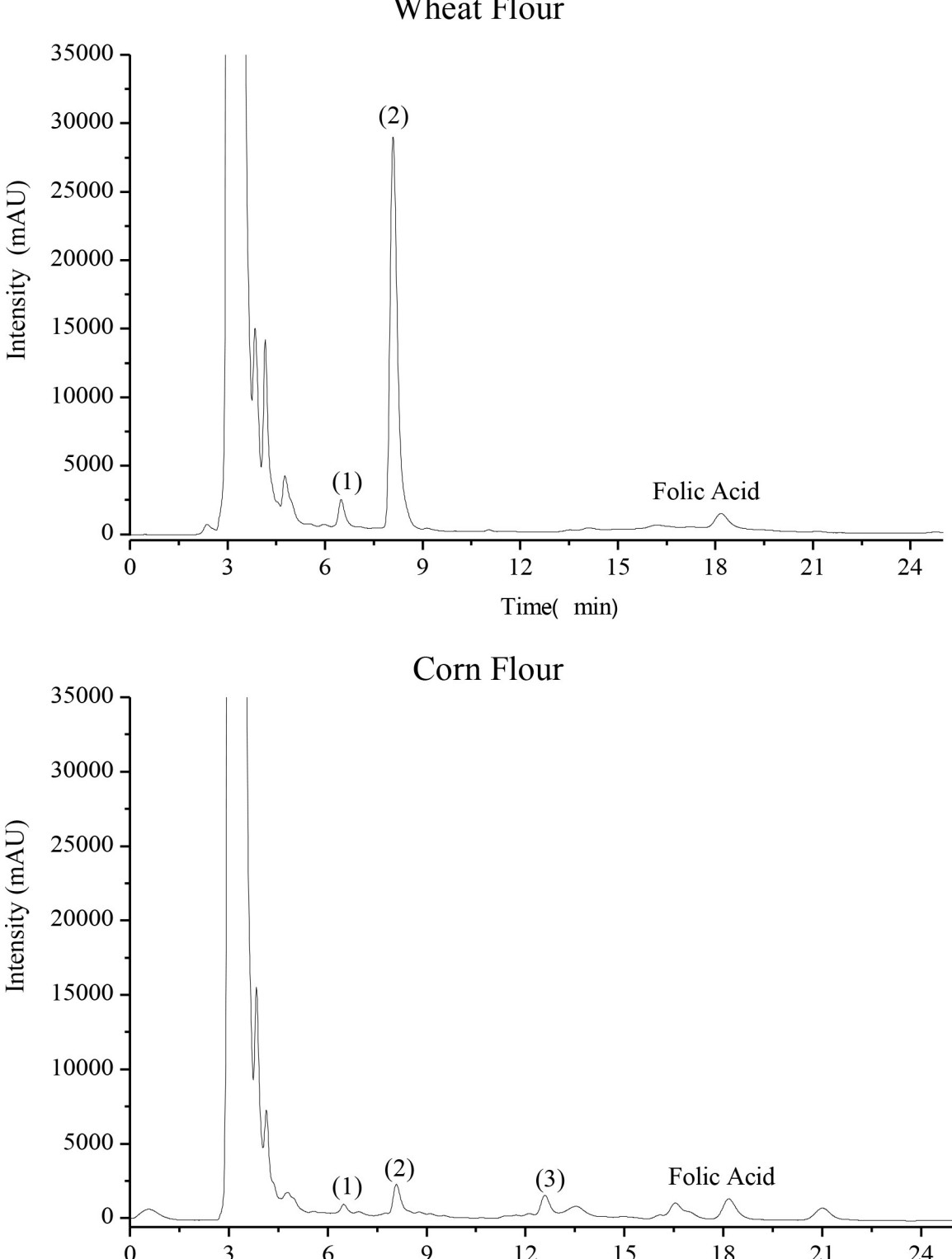

**Fig 2. Chromatograms of the laboratory fortified wheat and corn flours.** Chromatograms referring to the samples of wheat and corn flour laboratory fortified with folic acid.

folic acid content found has shown to be inferior to the one added to the farinaceous matrices through the fortification protocol (item 2). Although we performed two test procedures for homogenization, the first being more efficient due to a shorter exposure time of the material to environmental codes, the recovery rate was still between 51 and 53%.

There is a high variation in the folic acid content in commercialized corn and wheat flours and this may occur, probably, due to problems during the homogenization process, which can be related to the tendency the vitamin has to agglomerate, making its dispersion more difficult [15, 16, 17]. The significant variation in the folic acid content of the laboratory fortified corn and wheat flours may be related to the homogenization process performed on the workbench. In the present study, the time (12 minutes) used to mix the vitamin with the flour, and the equipment used for the homogenization (cutter) may have contributed to the low recovery of the added vitamin as the contents found after the fortification process, when compared to the added value, were lower for both types of flours. Studies evaluating folic acid content in Brazilian corn and wheat flours show that the amount of the vitamin varies considerably from one region to another.

Soeiro et al. [18], when investigating the quality of fortified flours sold in the state of São Paulo, found that the folic acid concentration in wheat flour was close to the value determined by the Brazilian legislation (150 µg.100g$^{-1}$), however, the corn flours had a much higher content (twice as much) than the one established. In a study carried out by Alaburda et al. [19] the results for the analysis of 33 samples of fortified wheat flour showed that 17 of them (51%) had concentration below the minimum level stated by the Brazilian legislation (1.50 µg.g$^{-1}$).

The quality of commercially fortified flours has been evaluated in other countries around the world and, therefore, a couple of papers on folic acid content have been published. Chandra-hioe, Bucknalll, Arcot [20] found that folic acid levels in commercially fortified flours were below the average concentrations (200µg.100g$^{-1}$ of flour) required by the Australian legislation. For flours sold in supermarkets in Poland, the average value reported for folic acid concentration was 237 µg.100g$^{-1}$ [21].

When analyzing commonly commercialized flours in the United States, whose fortification regulation stipulates a minimum of 140 µg.100g$^{-1}$, Rader et al. [22] found folic acid levels ranging from 33 µg.100g$^{-1}$ to 229 µg.100g$^{-1}$. Arcot, Shrestha and Gusanov et al. [23] reported values ranging from 82 µg.100g$^{-1}$ to 95 µg.100g$^{-1}$ in the same food matrix also commercialized in the United States.

In the preparations made with wheat flour, the folic acid content found in the raw mixture varied from 218 µg.100g$^{-1}$ to 431 µg.100g$^{-1}$. The lowest content was found in the white cream sauce and the highest content was found in the bread loaf. In the corn flour preparations, the folic acid content ranged from 272 µg.100g$^{-1}$ in the corn cake to 310 µg.100g$^{-1}$ in couscous, as it can be seen in Table 2.

The results for the true retention of folic acid in the preparations of wheat flour cake, white cream sauce, bread loaf, corn cake, and couscous after cooking are shown in Table 3. In the food preparations formulated with fortified wheat flour, the highest retention in folic acid content was observed for the bread loaf (87%), followed by the wheat flour cake (80%) and the white cream sauce (57%). In the preparations made with laboratory fortified corn flour, vitamin retention was high, being the highest value observed for the corn cake (99%) followed by the *couscous* (97%).

Fig 3 shows statistical analysis of variance (ANOVA), a posteriori test mean (+- 95% confidence interval), Dunnet with the value of p <0.0001, applied to fortified wheat flour and its preparations.

**Table 1. Content of folic acid in laboratory fortified.**

| Samples | μg.100g$^{-1}$ of folic acid ± SD | Similarity with the folic acid standard | Peak Purity | Analytical parameters | Calibration curve |
|---|---|---|---|---|---|
| WF | 487 ± 0,5 | 1,00 | 1,00 | LOD—230,56 | y = 4389.1x - 869.42 |
| CF | 474 ± 0,2 | 0,99 | 1,00 | LOQ—698,66 | R$^2$ = 0.9996 |

WF - wheat flour; CF - corn flour; SD - standard deviation. LOD and LOQ

n = 5

Samples 4 and 5 do not differ from each other not even when compared to the control (1—fortified wheat flour). Samples 2 and 3 also do not differ from each other, but they differ from the control (1) and samples (4) and (5).

Fig 4 shows the statistical analysis of variance (ANOVA), mean (± 95% confidence interval), Dunnet a posteriori test with the value of p <0.000, applied for evaluation of the folic acid content between the fortified corn flour and its preparations. Comparing to the control, sample 12, all other samples (8, 9, 10 and 11) showed a significant difference in folic acid content. Samples 8 and 9, 10 and 11 showed no significant difference in folic acid content: when compared to the raw mixture and after cooking of the same preparation and also when comparing the samples with each other.

Fig 5 shows the statistical treatment, T-Test (Kruskal-Wallis) for independent samples (t = 6.42; gl = 33; p <0.00001), which was applied for the evaluation of the folic acid content among the preparations considering the heat treatment used; dry-heat (1) and moist-heat (2).

The comparison between the two cooking methods let the authors conclude that there was a significant statistical difference in the content of folic acid in the preparations due to the cooking method. The preparations cooked using the dry-heat method (1) showed a higher concentration of folic acid as compared to the preparation cooked using the moist-heat method (2) which had the lowest content of the retained vitamin.

The differences in folic acid concentration in the different raw preparations (Table 2) compared to the amount of folic acid added to the farinaceous matrices used in the preparations suggest that the food preparation itself contributes to losses in the vitamin content. These observations were rectified when the statistical treatment applied (Figs 3 and 4) showed significant differences in the folic acid content between the control (fortified flour) and its preparations.

The results for the true retention of folic acid in the preparations of wheat flour cake, white cream sauce, bread loaf, corn cake and couscous after cooking are shown in Table 3.

The higher retention of folic acid in the preparations (Table 3) submitted to the dry-heat cooking method suggests that this technique is more appropriate when you want to effectively

**Table 2. Content of folic acid in the raw preparations.**

| Farinaceous matrix | Preparations | The Folic acid content in a raw preparation (μg.100g$^{-1}$) ± SD |
|---|---|---|
| Wheat flour | Cake | 335 ± 0,3 |
| | White Cream Sauce | 218 ± 0,2 |
| | Bread Loaf | 431 ± 0,4 |
| Corn flour | Corn cake | 272 ± 0,4 |
| | Couscous | 310 ± 0,3 |

SD–Standard deviation.

n = 5

**Table 3. True retention of folic acid in food preparations made with wheat and corn flour fortified in the laboratory.**

| Samples | Cooking conditions | | The Folic acid content in cooked preparation (mcg/g) ± SD | The Folic acid content in cooked preparation (g/100g) ± SD | Weight of cooked preparation (g) | The Folic acid content in a raw preparation (mcg/g) ± SD | Weight of the raw preparation (g) | RR (%) ± SD |
|---|---|---|---|---|---|---|---|---|
| | Time (min) | Temp. (˚C) | | | | | | |
| Wheat flour cake | 40 | 180 | 3,0 ± 0,001 | 0,30 ± 0,057 | 516 | 3,4 ± 0,0003 | 576 | 80 ± 17 |
| Bread loaf | 40 | 180 | 4,3 ± 0,0004 | 0,43 ± 0,032 | 335 | 4,3 ± 0,0004 | 383 | 87 ± 8 |
| White cream sauce | 5 | 100–60 | 2,3 ± 0,001 | 0,23 ± 0,008 | 117 | 2,2 ± 0,0001 | 218,42 | 57 ± 3 |
| Corn cake | 40 | 180 | 2,9 ± 0,001 | 0,29 ± 0,038 | 511 | 2,7 ± 0,0004 | 546 | 99 ± 9 |
| Couscous | 5 | 100–40 | 3,1 ± 0,0002 | 0,31 ± 0,011 | 250 | 3,1 ± 0,0003 | 260 | 97 ± 11 |

RR - real retention; SD - standard deviation. n = 5.

retain folic acid in the prepared food. Samples cooked using the dry-heat method (wheat flour cake, bread loaf, couscous and corn cake) were statistically higher in the folic acid content as compared to the preparation cooked using moist-heat (white cream sauce) (Fig 5). The highest losses were observed in the white cream sauce (43%), and even before heat treatment, it showed the lowest concentration of folic acid suggesting that the direct contact of the food with the cooking water promotes considerable losses in the content of this vitamin that continues to be lost during cooking.

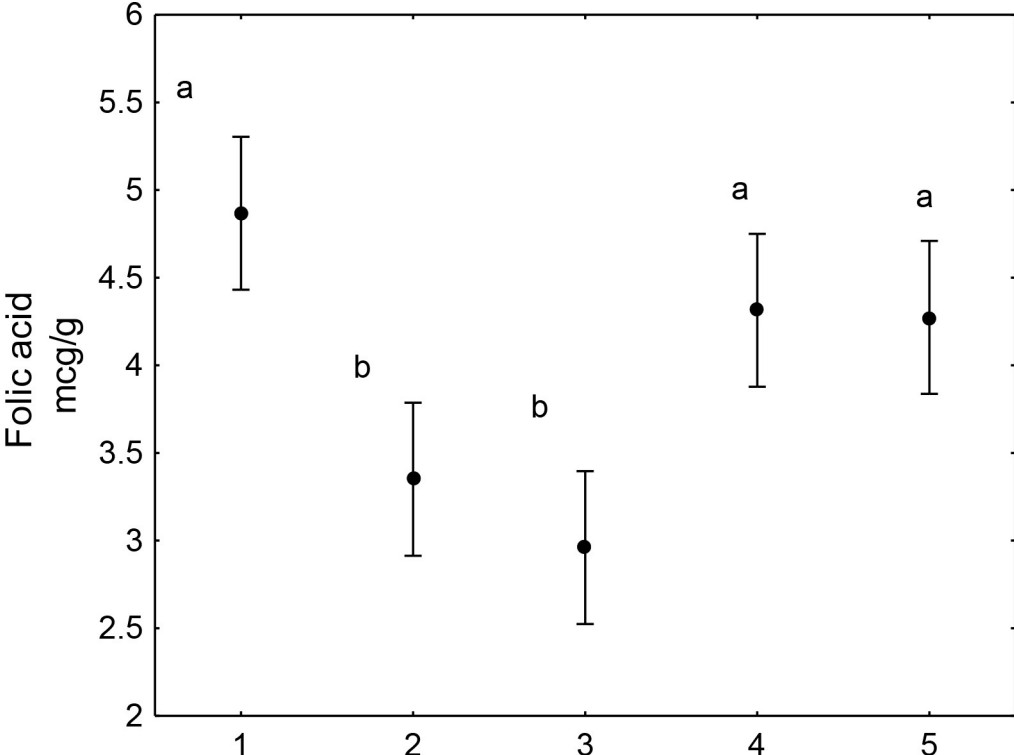

**Fig 3. Comparison between treatments of wheat flour fortified with folic acid.** Different letters indicate differences between means (p <0.0001). 1—folic acid fortified wheat flour (1,000μg.100g-1); 2—wheat flour cake before cooking; 3—wheat flour cake after cooking; 4 –bread dough before cooking; 5 –bread loaf after cooking.

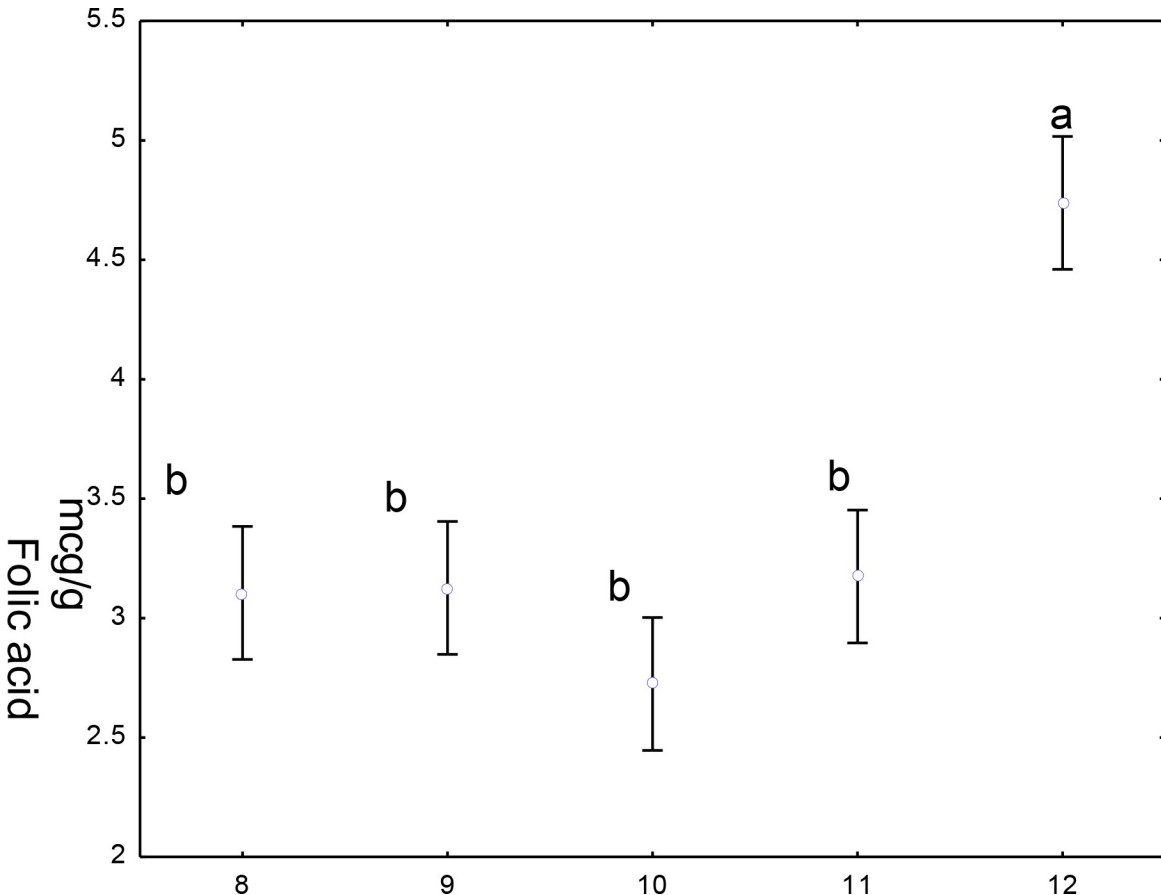

**Fig 4. Comparison between treatments of corn flour fortified with folic acid.** Different letters indicate differences between means (p <0.0001). 8—*couscous* before cooking; 9 *couscous* after cooking; 10—corn cake before cooking; 11—corn cake after cooking; 12—corn flour fortified with folic acid.

The decrease in the folic acid content after cooking by immersion in water can be explained by the water-soluble nature of the vitamin that contributes to the losses occurring mainly due to the leaching to the cooking water in food preparations. However, in this case, it is more likely that the loss of folic acid in the white cream sauce is related to the hydrolysis of the vitamin in the cooked food matrix than to leaching, since the loss of the vitamin to the cooking water is unlikely as the water remained incorporated into the food preparation while it was being made and even after the cooking process was finished.

The variation in folic acid concentration in the food preparations made with wheat flour (Fig 4) also showed a relationship with the cooking method used; however, it is further believed that the preparation stage of the mixture can also contribute to a higher or lower concentration of the vitamin. The observed folic acid retention for bread loaf (87%) was higher when compared to the wheat flour cake (80%) and white cream sauce (57%). When comparing the wheat flour cake and bread loaf preparations, it is observed that both were submitted to the same heat treatment conditions (180˚ C / 40 minutes), however, the preparation method used for the bread dough, which uses the mechanical energy passed on to the dough during the kneading to induce the breakage and formation of bonds, results in the formation of a gluten network [24]. This network plays an important role, in affecting the water absorption capacity (which, in this case, is higher), cohesion, viscosity, and elasticity of the dough [25].

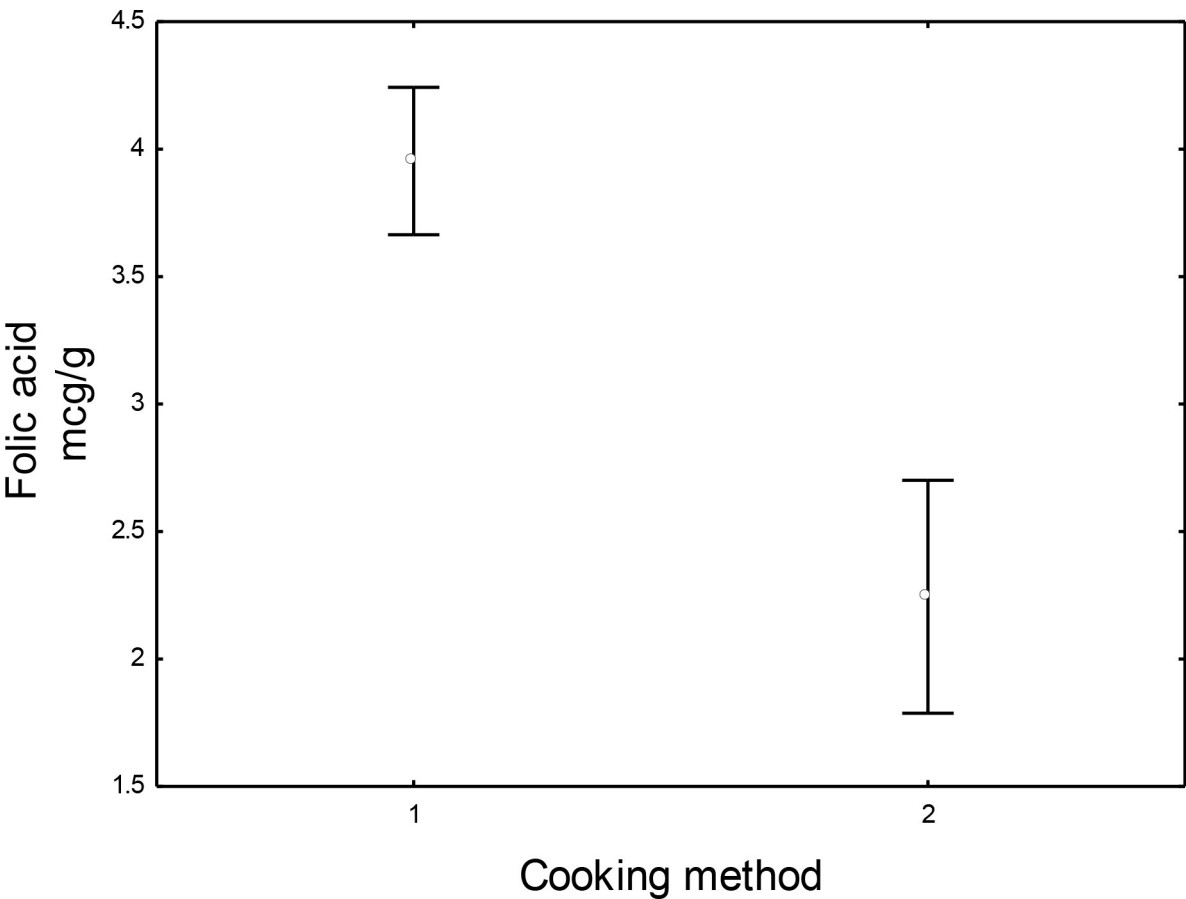

**Fig 5. Comparison between dry-heat and moist-heat cooked preparations 1 –cooking using dry-heat; 2—cooking using moist heat.**

It is believed that the formation of the gluten network in the bread loaf, which also happens on the wheat flour cake (in a smaller degree), had an influence in the retention of folic acid in the food matrix, being the retention higher on the bread loaf due to the greater mechanical work applied when preparing the dough that was, therefore able to absorb more water than the cake and thus have a better connection of the vitamin with the water that was retained in the gluten network formed. Folic acid has a water-soluble nature and, if during the cooking process by immersion in water it is lost by being diffused in the cooking water, in the case of bread and wheat cake, it can be retained by the water present in the gluten network.

Regarding the preparations made with fortified corn flour, both preparations, corn cake and couscous, showed the best folic acid retentions in the present study. The statistical test (Fig 5) showed that the differences between the two preparations were not significant and, therefore, there were no significant losses of folic acid.

The higher folic acid retention in the corn flour cake can be explained by the formation of a protein network, similar to the one found on the bread loaf and the wheat flour cake, which allows for there to be an interaction between the network components and the folic acid, entrapping the vitamin in the food matrix. The protein network observed in the corn flour preparations is formed by the Zein protein which, when exposed to temperatures higher than 35˚C, loses its native state, turning into a new structural arrangement with structural changes similar to the ones observed in the viscoelastic polymers of gluten [26].

According to Penalva et al. [9] there is no information that suggests the presence of protein-ligand binding sites for the folic acid with the Zein, however, according to the results of the study involving the creation of Zein nanoparticles for the oral delivery of folic acid, the authors suggest that a fraction of the folic acid was stabilized inside the Zein protein matrix by the non-covalent bonds. Therefore, the higher folic acid retention obtained in the corn flour cake can be explained by the interaction between the folic acid with the hydrophobic aminoacids of the Zein which prevents the oxidation of the vitamin.

The evaluation of folic acid stability in preparations developed with enriched flours was also studied by Phillips [27]. When evaluating tortillas and chip tortillas made from fortified cornmeal, the author found a 13% loss of folic acid. in cooked tortillas, but no loss during frying leading him to conclude that folic acid is relatively stable to the dry heat cooking process.

In Ireland, a study conducted to quantify folic acid reduction during baking of four different types of commercial breads [16] found that the thermal degradation of folic acid was between 21.9% and 32, 1%, is the percentage of vitamin degradation similar for white bread, baguettes, and brown soda and significantly higher for whole-grain bread when compared to other types of bread tested.

In another study by Silveira et al. [28] the content and stability of folic acid were verified in fortified rice after four different cooking methods (frying, boiling, microwave and boiling in food service). After analysis the authors concluded that the lowest folic acid content (0.17mg / kg) was found in food service rice and the highest retention (96.11%) in fried rice.

In spite of the limitation observed in paper availability concerning the subject, it is acknowledged that the methods that best preserve the content of folic acid are those that use dry-heat, since the moist-heat methods, especially those that immerse the food in the cooking water, contribute to higher losses. In this way, the use of cooking techniques that better preserve the folic acid content in foods, or even the ingestion of the cooking water from the food prepared by immersion, contribute to its greater preservation/retention in cooked food and, consequently, an adequate ingestion of the required amount of the vitamin, thus avoiding the manifestation of pathologies due to its deficiency or low intake.

Food fortification has to be considered as an important tool for the population to reach its minimum nutrient requirements as recommended in food policy interventions [19], however, the monitoring and controlling of the amount of the micronutrient added to foods that go through this technological process are of extreme importance to guarantee the commercialization of these products containing the actual amount of micronutrients required by legislation and stated on the food labels.

It should also be emphasized that enriched flours are used as ingredients for the preparation of different foods and that, during food preparation for consumption, vitamin losses may occur due to degradation or leaching of the vitamin, which will have a direct impact on the consumption of the added micronutrient [18].

Therefore, good control of how micronutrients are added in the final food products is necessary to guarantee the quality of processed foods and for the control and regulation of public policies [19]. Although the determination of the folic acid content in fortified wheat and corn flours has a significant effect on the practice of fortification performed by industries worldwide, it has received little attention in Brazil.

The results reported in the present study provide important information about the practice of folic acid fortification of wheat and corn flours carried out in Brazilian industries, especially those flours commercialized in the Northeast of the country, contributing to reinforcing the necessity of the Brazilian authorities to better monitor the fortification process of farinaceous matrices.

## 5. Conclusions

Commercially fortified wheat and corn flours showed the absence of industrially added folic acid. That is believed to happen due to the homogenization process carried out in the industries. The results reported show a need to monitor the folic acid fortification programs of farinaceous matrices, especially when considering no detection of the vitamin in all the food products analysed.

Different cooking techniques cause significant losses of folic acid, but these losses are related to the complexity of the food matrix as well as the cooking method applied. Additional studies investigating the stability of folic acid in other preparations commonly consumed by the population and using various cooking methods are necessary not only to improve the techniques in order to increase the folic acid retention in the cooked food to be consumed, but also to update the food composition databases and thus provide further useful information for calculations of dietary intake and nutritional adequacy.

The information on the stability of folic acid in fortified foods that have been submitted to different cooking techniques is limited and the majority of the studies are focused on investigating the influence of different cooking methods under the folate content in vegetables. Therefore, there is a need for more research on evaluations of the stability of folic acid in foods prepared from fortified matrices, especially wheat and corn flours that are commonly used as ingredients for the preparation of different foods regularly consumed by the population.

## Author Contributions

**Conceptualization:** Emmanuela Prado de Paiva Azevedo, Samuel de Santana Khan, Carlos Bôa-Viagem Rabelo, Ana Carolina dos Santos Costa, Margarida Angélica da Silva Vasconcelos.

**Formal analysis:** Emmanuela Prado de Paiva Azevedo, José Roberto Botelho de Souza.

**Funding acquisition:** Beate Saegesser Santos.

**Investigation:** Eryka Maria dos Santos Alves.

**Methodology:** Eryka Maria dos Santos Alves, Leonardo dos Santos Silva, José Roberto Botelho de Souza, Beate Saegesser Santos.

**Project administration:** Eryka Maria dos Santos Alves, Beate Saegesser Santos, Carlos Bôa-Viagem Rabelo, Ana Carolina dos Santos Costa, Margarida Angélica da Silva Vasconcelos.

**Resources:** Eryka Maria dos Santos Alves, Samuel de Santana Khan.

**Supervision:** Emmanuela Prado de Paiva Azevedo, Beate Saegesser Santos, Clayton Anderson de Azevedo Filho.

**Validation:** Clayton Anderson de Azevedo Filho.

**Writing – original draft:** Eryka Maria dos Santos Alves, Samuel de Santana Khan.

**Writing – review & editing:** Emmanuela Prado de Paiva Azevedo, Samuel de Santana Khan, Clayton Anderson de Azevedo Filho, Margarida Angélica da Silva Vasconcelos.

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
