## [Decision Letter · Decision Letter 0]

15 Oct 2019

PONE-D-19-26781

Determination of folic acid by HPLC in fortified wheat and corn flour and evaluation of its retention in preparations submitted to different cooking methods

PLOS ONE

Dear Dra. AZEVEDO,

Thank you for submitting your manuscript to PLOS ONE. After careful consideration, we feel that it has merit but does not fully meet PLOS ONE’s publication criteria as it currently stands. Therefore, we invite you to submit a revised version of the manuscript that addresses the points raised during the review process.

We would appreciate receiving your revised manuscript by Nov 29 2019 11:59PM. To enhance the reproducibility of your results, we recommend that if applicable you deposit your laboratory protocols in protocols.io, where a protocol can be assigned its own identifier (DOI) such that it can be cited independently in the future. For instructions see: http://journals.plos.org/plosone/s/submission-guidelines#loc-laboratory-protocols

We look forward to receiving your revised manuscript.

Kind regards,

Walid Elfalleh, Ph.D

Academic Editor

PLOS ONE

Journal Requirements:

3.  Please include in your Methods section vendor information about the ingredients used and the flours tested, in order to ensure reproducibility of the results by other researchers. 2

4.  Please consider whether the graphs in figures 3 and 4 should have the data points connected by lines or not.

5.  Thank you for stating the following financial disclosure:  "The funders had no role in study design, data collection and analysis, decision to publish, or preparation of the manuscript."

Please provide an amended Funding Statement that declares *all* the funding or sources of support received during this specific study (whether external or internal to your organization) as detailed online in our guide for authors at http://journals.plos.org/plosone/s/submit-now.  Please state what role the funders took in the study.  If any authors received a salary from any of your funders, please state which authors and which funder. If the funders had no role, please state: "The funders had no role in study design, data collection and analysis, decision to publish, or preparation of the manuscript."

6.  Please amend either the title on the online submission form (via Edit Submission) or the title in the manuscript so that they are identical.

7.  Please amend either the abstract on the online submission form (via Edit Submission) or the abstract in the manuscript so that they are identical.

Additional Editor Comments (if provided):

I have completed my evaluation of your manuscript. Although I am recommending revision, you have a major task to radically improve the content of your paper and its organization and interpretation. Each of the reviewer comments need to be addressed individually and listed. The paper is really a borderline for rejection at this stage.

other comments:

The statistical analysis is missed in different illustrations (Table 2 and Table3).

The manuscript includes several grammar and syntax errors and should be revised by a specialist.

Reviewers' comments:

Reviewer's Responses to Questions

**Comments to the Author**

1. Is the manuscript technically sound, and do the data support the conclusions?

Reviewer #1: Partly

Reviewer #2: Yes

Reviewer #3: Partly

Reviewer #4: Yes

Reviewer #5: Partly

Reviewer #6: No

Reviewer #7: Yes

Reviewer #8: Partly

2. Has the statistical analysis been performed appropriately and rigorously? 

Reviewer #1: I Don't Know

Reviewer #2: Yes

Reviewer #3: I Don't Know

Reviewer #4: Yes

Reviewer #5: I Don't Know

Reviewer #6: Yes

Reviewer #7: Yes

Reviewer #8: Yes

3. Have the authors made all data underlying the findings in their manuscript fully available?

Reviewer #1: Yes

Reviewer #2: Yes

Reviewer #3: Yes

Reviewer #4: Yes

Reviewer #5: Yes

Reviewer #6: Yes

Reviewer #7: Yes

Reviewer #8: Yes

4. Is the manuscript presented in an intelligible fashion and written in standard English?

Reviewer #1: Yes

Reviewer #2: Yes

Reviewer #3: Yes

Reviewer #4: Yes

Reviewer #5: No

Reviewer #6: No

Reviewer #7: Yes

Reviewer #8: Yes

5. Review Comments to the Author

Reviewer #1: • The Title: clearly describe the article

• Abstract: it reflect the content of the article

it describe what the author hoped to achieve accurately, and clearly state the problem being investigated and summarize relevant research to provide context, and explain what other authors’ findings, It describe the experiment, the hypothesis(es) and the general experimental design or method.

The author explain how the data was collected and analyzed there is sufficient information present for the readers, the article identify the procedures the method are advanced and new.

• The author/s explain in words what he discovered in the research. It should be clearly laid out and in a logical sequence. the statistics are correct

• Conclusion/Discussion: the claims in this section supported by the results, they seem reasonable, the authors indicated how the results relate to expectations and to earlier research.

It is well integrated with current research

• Language: Article is good written

• The Tables and figures describe the data accurately

Accepted

Reviewer #2: it is my pleasure to review the manuscript entitles "Determination of folic acid by HPLC in fortified wheat and corn flour and evaluation of its retention in preparations submitted to different cooking methods", this paper should become acceptable for publication. So my decision is accept the publication.

Reviewer #3: 1-Line 54 till 55:(Your low ingestion) not giving a clear meaning that lack of folic acid or its deficiency can cause series diseases, I recommend re-writing these lines.

2-page 9 in Identification and quantification:The authors didn't mention the detector type, is it Diode array or UV detector to perform such spectral similarities.

3-Page 13 (line 326) the word maize has to be corrected to corn.

4- Page 20 (line 437): biting site has to be corrected as binding site.

5-In Fig. 3 & 4 in the supporting information: you have to name the Y-axis.

Reviewer #4: Reviewer Comment

EMMANUELA AZEVEDO and his coauthors wrote a manuscript about their investigation on Determination of folic acid by HPLC in fortified wheat and corn flour and evaluation of its retention in preparations submitted to different cooking methods. The above mentioned goal has been achieved in a clear way. However, the manuscript requires revision though it may eventually be publishable in this journal.

The manuscript needs English editing; some sentences are not clear due to mistakes (e.g. punctuation, grammatical or typos errors).

General comments:

• The authors must follow the formatting of the Journal as indicated in the authors guidelines.

This implies for reference formatting and numbering of sections of the manuscript.

• The authors should further compare their results with other works on fortified wheat and corn flour

• Another method of preparation using moist heat should have been added to strengthen your results not just only one.

• Tables and figures have to be self-explanatory. All abbreviations have to be explained in each table/figure.

Specific comments:

In the whole manuscript: μg.mL-1 must be changed to μg mL-1 (without a dot inbetween and -1 should be superscript) or change to μg/mL

Why do you change from μg to g and not to mg as in line 106 and other places ‘Approximately 5,000 μg (0.0050 g) of folic acid’??? why not be “Approximately 5,000 μg (5 mg) of folic acid”…etc.

Line 38 add ‘to’ after ‘submitted’

Line 44 remove the before zein and make it capital ‘Zein’

Line 45 remove ‘of’ before 87% and change ‘the on bread’ to ‘on the bread’

Line 47 change ‘the mixture’ to ‘mixing

Line 54: remove ‘Your’

Line 75 and 76: remove ‘the’ before ‘the method’ and ‘the cooking time’.

Line 159: add a space between the 8 and g in ‘8 g’

Line 205: what is the meaning of ‘Then the samples were centrifuged in 5000 g rotation’ is it ‘Then the samples were centrifuged at 5000 rpm rotation’

Line 256: ‘The methodology applied allowed for there to be retention of folic acid’ change ‘for there to be’ to ‘for the’.

Line 257: why was 290 nm used for measuring the absorptivity and not 280 nm, was it in the literature.

Line 268: Fig. 1 Chromatogram o commercially fortified wheat flour and corn flour. I think this ‘o’ is ‘of’

In Fig. 3 and 4 what does mcg/g stand for, if it is microgram then it has to be written as ‘μg/g’.

There were newer versions of ICH guidelines than 1994 please consider.

• It is better to switch the paragraph starting at line 444-450 to the end of the discussion.

The manuscript covers an interesting topic with interesting results. I would recommend this manuscript for publication after major revision.

Reviewer #5: please find it also as the attached file for author

Dear authors

Please find the following suggestions and I hope to enhance your paper for good publication ‎

Introduction ‎

‎1-‎ Please highlight the important of folic acid for the human health and the negative effect ‎of its deficiency to spotlight the important of your work ‎

‎2-‎ Please highlight‎ some of the previous reports discussed thermal effect on the retention of ‎folic acid and other micronutrients through the different cooking methods

Material and methods

‎1.‎ Samples and reagents

‎1-‎ Lines (88-90): Commercially fortified wheat and corn flours and laboratory fortified ‎wheat and corn flours were evaluated in the laboratory. (what the added value of this ‎sentence please delete it)‎

‎2-‎ ‎Lines (91-92): ‎ different brands of wheat flour (WF); different brands of corn flour (CF) ‎‎(please add the name of used brands).‎

‎3-‎ Lines (95): ‎item 2.2. (there is no 2.2. please indicate it as 2)‎

‎2.‎ Protocol for laboratory folic acid fortification of wheat and corn flour

‎1-‎ Lines (108): ‎a concentration of 1000 μg of folic acid for every 100 g of wheat or corn ‎flour. (Why you exceeded the Brazilian recommended dose; a maximum limit (220 ‎μg.100g-1) for addition of the vitamin in order to avoid exposure of the population to an ‎unnecessary risk (BRAZIL, 2017). ‎

‎2-‎ ‎Lines (109-110): ‎two procedures of homogenization were used (I recommended the first ‎procedure only because it is the simplest and applicable one either for individual or ‎industrial use, so please delete the second procedure).‎

‎3-‎ ‎ Lines (124-127): ‎At the end of the fortification procedure, the total time used for the ‎homogenization of the vitamin into the farinaceous matrix, including manual (1 min for ‎each 100 g) and mechanical homogenization (1 min for every 100 g + 2 minutes for 500 ‎g), was of 12 minutes. (Why you mentioned this sentence to compare the spent time and ‎what about the multiple steps and tools is it ok for industrial application??)‎

‎4-‎ Lines (128): ‎packages to prevent degradation (please describe the full conditions (temp. ‎container material and color, humidity, etc...) you used to prevent degradation ‎ for ‎storage until used).‎

‎3.‎ Food preparations

‎1-‎ Lines (133): ‎laboratory folic acid fortified wheat flour and corn flour, (which one you ‎used? Homogenized with procedure 1 or 2).‎

‎4.1. Sample extraction‎

‎1- Lines (200-201): ‎The methodological procedure was a solid-liquid extraction developed ‎and used for plant matrices by Paiva et al. (PAIVA et al., 2012). (Please replace it to; The ‎folic acid was extracted from the samples (fortified wheat flour and corn flour, raw ‎preparations, and cooked preparations) following the methods of PAIVA et al. (2012).‎

‎2-‎ Lines (205): ‎ in 5000 g rotation (please replace in with at)‎

‎4.2. Chromatographic conditions‎

‎1-‎ Please indicate the full description of HPLC equipment used in this analysis with the ‎country of made, the type of the detector used and wavelength at which the folic acid ‎was detected. Also, indicate the type and specification of the used column.‎

‎2-‎ Line (218); The column was washed with water and conditioned with acetonitrile ‎‎(Merck-grade HPLC) at the end of the analyses (PAIVA et al., 2012). (Please delete ‎this sentence and move the reference in the right place) ‎

‎4.3.Identification and quantification‎

‎1-‎ ‎(1000 μg.mL-1 folic acid) (please replace it with (1000 μg.mL-1 folic acid)‎

‎5.True retention ‎

‎(MURPHY; CRINER, GRAY, 1975) (Please correct it to Murphy‎‏ ‏et al‏., 1975)‏

Results and discussion

‎1-‎ ‎Line (256-257);‎The methodology applied allowed for there to be retention of folic acid in ‎‎18.2 minutes at maximum absorptivity in 290nm. (Please delete it because you mentioned ‎it in the methodology section).‎

‎2-‎ Lines (265-266);‎‏ ‏‎1. Determination of folic acid in commercially fortified and laboratory ‎fortified wheat and corn flours‎‏ ‏‎(please delete it) ‎

‎3-‎ ‎‏ ‏Line (268); ‎Fig. 1 Chromatogram o commercially fortified wheat flour and corn flour. ‎‎(Please make it: HPLC chromatogram of commercially fortified wheat flour and corn ‎flour evaluated in July (A) and September (B)).‎

‎4-‎ ‎‏ ‏Lines (269-274);‎ ‎ Chromatograms for one of the samples of commercially fortified wheat ‎flour and corn flour evaluated in July (A) and one of the wheat flour and corn flour ‎samples evaluated in September (B) Chromatographic conditions: C18 (150 x 4.6 mm ‎‎5μm) chromatographic column (Allcrom Phenomenex) with the oven at 40 °C, isocratic ‎elution using potassium phosphate buffer (pH 2.0) and methanol (85:15) as mobile phase ‎at a flow of 0.5 mL/min-1 and UV detection with wavelength of 290 nm. (Please delete it ‎and mentioned it in the methodology section‎‏ ‏as general condition for all HPLC analysis‏), ‏and apply it for the other figures‏.‏

‎5-‎ Lines (276);‎ (1.000μg.100g-1) ( Please correct it to ‎1000μg.100g-1‎‏) ‏and check it through ‎the paper‏ ‏

‎6-‎ ‎ the presentation of the results is not acceptable because of:‎

‎ 1- Using different units for folic acid in different tables and figures

‎2- Content of folic acid in the raw preparations in table 1 is different than in table 3 for ‎example for Bread Loaf is 431 ± 0,4(μg.100g-1) ± SD in table 1 and is 0,43 ± 0,030 (g/100g) ‎‎± DP even 431μg equal 0.000431‬g and what is the meaning of DP. ‎

‎3- Repeated data on the tables and figures, figures 3, 4 derived from table 3, so table 3 is ‎enough

‎4- Table 1 contains extra statistical data have no effect on the study

‎5- Cooking conditions columns in Table 3 did not give any information as it as the same ‎mentioned in materials and methods and are not variable studied conditions ‎

‎7-‎ The folic acid peak in figure 2 is not strong peak for trusted quantification of folic acid as ‎the other peaks in the chromatogram are at so near or at the same height. (I highly ‎recommended the preparation of standard curve for folic acid inside each raw ‎preparation. After that, you can use this standard curve for quantification of folic acid ‎inside the raw and cooked preparations. ‎

‎8-‎ Using of standard curve for folic acid inside each raw preparation will verify the retention ‎time for folic acid on the chromatogram and will give you the recovery percent of the ‎solvent used for folic acid extraction from the preparation and it will answer you about ‎the big variation on the folic acid quantities on your different preparations as the ‎extraction method has a strong effect on the recovery content of folic acid. ‎

‎9-‎ The interpretation of discussion needs more supporting results for discussing the ‎variable ‎results ‎against the different conditions of cooked preparation of temperature ‎and ‎foodstuff added per each ‎preparation. ‎ ‎

Reviewer #6: The MS needs depth revision and English editing. The aims of the study are clear, but the Introduction does not review in depth the problematic and the hypothesis of the MS. I think that the MS is not sufficiently novel and interesting to warrant publication. The work is not original.

Reviewer #7: The manuscript concerned with determination of Folic acid content in fortified wheat and corn flours and their folic acid

retention in food preparations by using cooking methods: baking, deep

frying and steaming.

In the corn flour preparations, Folic acid retention is high compared to Wheat flour preparations Why?

The author should do another experiment in which a mixture of folic acid fortified corn and wheat flours are mixed together to testify or investigate the effect of mixing corn with wheat in Foods on the stability or retention of folic acid.

Reviewer #8: The idea is good.

There is no matching research

The language of the manuscript is good with some simple mistakes.

The methods do not have technical errors that spoil the search.

But:

Failure to standardize the expression of concentrations. Makes it difficult to review the results

Data transformation has reduced the variance between treatments, which requires statistical re-analysis

This has led to a change in results but will not affect the overall direction of the findings or final conclusions.

There are some problems with references (in formulation and completeness only).

6. PLOS authors have the option to publish the peer review history of their article (what does this mean?). If published, this will include your full peer review and any attached files.

Reviewer #1: No

Reviewer #2: Yes: Doha H Abou Baker

Reviewer #3: No

Reviewer #4: Yes: Marwa Y. Issa

Reviewer #5: No

Reviewer #6: No

Reviewer #7: No

Reviewer #8: Yes: Sabry A. AbdAllah

Lecturer of pesticides chemistry and toxicology

plant protection department, Faculty of Agriculture

---

## [Author Response · Author response to Decision Letter 0]

3 Jan 2020

Dear editorial below I send as replies of the comments and suggestions of the reviewers. For each comment below follows an answer. We appreciate all the considerations you have made and we seek the ultmost from all that has been requested.

All line locations were made by the document "Revised manuscript with track changes"

Answer: The manuscript was reviewed by an expert in the English language.

Answer: We fulfill this request.

Answer: We fulfill this request.

Answer: We fulfill this request.

3. Please include in your Methods section vendor information about the ingredients used and the flours tested, in order to ensure reproducibility of the results by other researchers. 

Answer: We fulfill this request. Line 96 to 98

4. Please consider whether the graphs in figures 3 and 4 should have the data points connected by lines or not.

Answer: In fact the points should not be linked as the treatments are independent. Figures 3 and 4 have been changed.

5. Thank you for stating the following financial disclosure: "The funders had no role in study design, data collection and analysis, decision to publish, or preparation of the manuscript."

6. Please amend either the title on the online submission form (via Edit Submission) or the title in the manuscript so that they are identical.

Answer: Was made

7. Please amend either the abstract on the online submission form (via Edit Submission) or the abstract in the manuscript so that they are identical.

Answer: Was made

Additional Editor Comments (if provided):

I have completed my evaluation of your manuscript. Although I am recommending revision, you have a major task to radically improve the content of your paper and its organization and interpretation. Each of the reviewer comments need to be addressed individually and listed. The paper is really a borderline for rejection at this stage.

Answer: We appreciate that you have considered this manuscript for further revision.

other comments:

The statistical analysis is missed in different illustrations (Table 2 and Table3).

Answer: For the construction of tables 2 and 3, only the mean and standard deviation evaluation was used.

The manuscript includes several grammar and syntax errors and should be revised by a specialist.

Answer: The manuscript was reviewed by an expert in the English language.

Reviewer #1: 

 The Title: clearly describe the article

• Abstract: it reflect the content of the article

it describe what the author hoped to achieve accurately, and clearly state the problem being investigated and summarize relevant research to provide context, and explain what other authors’ findings, It describe the experiment, the hypothesis(es) and the general experimental design or method.

The author explain how the data was collected and analyzed there is sufficient information present for the readers, the article identify the procedures the method are advanced and new.

• The author/s explain in words what he discovered in the research. It should be clearly laid out and in a logical sequence. the statistics are correct

• Conclusion/Discussion: the claims in this section supported by the results, they seem reasonable, the authors indicated how the results relate to expectations and to earlier research.

It is well integrated with current research

• Language: Article is good written

• The Tables and figures describe the data accurately

Accepted

Answer: Thank you for your considerations.

Reviewer #2:

 it is my pleasure to review the manuscript entitles "Determination of folic acid by HPLC in fortified wheat and corn flour and evaluation of its retention in preparations submitted to different cooking methods", this paper should become acceptable for publication. So my decision is accept the publication.

Answer: Thank you for your considerations.

Reviewer #3:

 1-Line 54 till 55:(Your low ingestion) not giving a clear meaning that lack of folic acid or its deficiency can cause series diseases, I recommend re-writing these lines.

Answer: The paragraph has been rewritten, check lines 60 through 62.

2-page 9 in Identification and quantification:The authors didn't mention the detector type, is it Diode array or UV detector to perform such spectral similarities.

Answer: All equipment information including the detector description has been entered in item 4.2. Chromatographic conditions, lines 224 to 229.

3-Page 13 (line 326) the word maize has to be corrected to corn.

Answer: Change made, check line 344

4- Page 20 (line 437): biting site has to be corrected as binding site.

Answer: Change made, check line 455 and 456

5-In Fig. 3 & 4 in the supporting information: you have to name the Y-axis.

Answer: Y axis has been named, check the changes in figures 3 and 4

Reviewer #4: 

Reviewer Comment

EMMANUELA AZEVEDO and his coauthors wrote a manuscript about their investigation on Determination of folic acid by HPLC in fortified wheat and corn flour and evaluation of its retention in preparations submitted to different cooking methods. The above mentioned goal has been achieved in a clear way. However, the manuscript requires revision though it may eventually be publishable in this journal.

The manuscript needs English editing; some sentences are not clear due to mistakes (e.g. punctuation, grammatical or typos errors).

Answer: The article was reviewed by an English language expert. We hope to have better manuscript quality

General comments:

• The authors must follow the formatting of the Journal as indicated in the authors guidelines.

This implies for reference formatting and numbering of sections of the manuscript.

Answer: We reviewed the article regarding the format of the magazine and met the two items mentioned.

The authors should further compare their results with other works on fortified wheat and corn flour

Answer: We seek to discuss our results using references that have also worked with these matrices, as can be seen from Lines 323 to 329.

Another method of preparation using moist heat should have been added to strengthen your results not just only one.

Answer: Answer: In fact, during a statistical assessment it is observed that the data are stronger if there are more matrices for heat humid. However, our findings have a significant value for the scarce information on cooked and prepared foods, thus maintaining the results of the co-study method in the study.

Tables and figures have to be self-explanatory. All abbreviations have to be explained in each table/figure.

Answer: The tables presented contain the values obtained in the analyzed material. Table 1 refers to the method parameters. We seek to address the discussion of results by focusing on the most relevant data and discussing them, so we hope that the reader can make better use of the material.

Specific comments:

In the whole manuscript: μg.mL-1 must be changed to μg mL-1 (without a dot inbetween and -1 should be superscript) or change to μg/mL

Answer: A requested correction has been made.

Why do you change from μg to g and not to mg as in line 106 and other places ‘Approximately 5,000 μg (0.0050 g) of folic acid’??? why not be “Approximately 5,000 μg (5 mg) of folic acid”…etc.

Answer: We seek to present the results also in grams because we are dealing with food prepared for consumption. In this sense, the reader could extrapolate a faster interpretation of nutrient availability in food per gram of food.

Line 38 add ‘to’ after ‘submitted’

Answer: A requested correction has been made.

Line 44 remove the before zein and make it capital ‘Zein’

Answer: A requested correction has been made.

Line 45 remove ‘of’ before 87% and change ‘the on bread’ to ‘on the bread’

Answer: A requested correction has been made.

Line 47 change ‘the mixture’ to ‘mixing

Answer: A requested correction has been made.

Line 54: remove ‘Your’

Answer: A requested correction has been made.

Line 75 and 76: remove ‘the’ before ‘the method’ and ‘the cooking time’.

Answer: A requested correction has been made.

Line 159: add a space between the 8 and g in ‘8 g’

Answer: A requested correction has been made.

Line 205: what is the meaning of ‘Then the samples were centrifuged in 5000 g rotation’ is it ‘Then the samples were centrifuged at 5000 rpm rotation’

Answer: Samples were centrifuged at 5000 g as described in line 218.

Line 256: ‘The methodology applied allowed for there to be retention of folic acid’ change ‘for there to be’ to ‘for the’.

Answer: A requested correction has been made.

Line 257: why was 290 nm used for measuring the absorptivity and not 280 nm, was it in the literature.

Answer: corrected to 280nm, it was a typo.

Line 268: Fig. 1 Chromatogram o commercially fortified wheat flour and corn flour. I think this ‘o’ is ‘of’

Answer: A requested correction has been made.

In Fig. 3 and 4 what does mcg/g stand for, if it is microgram then it has to be written as ‘μg/g’.

Answer: A requested correction has been made.

There were newer versions of ICH guidelines than 1994 please consider.

Answer: Answer: In fact there is a new version, but the one we had access to was 1994.

It is better to switch the paragraph starting at line 444-450 to the end of the discussion.

Answer: A requested correction has been made.

The manuscript covers an interesting topic with interesting results. I would recommend this manuscript for publication after major revision.

Answer: thank you for your considerations.

Reviewer #5: 

Dear authors

Please find the following suggestions and I hope to enhance your paper for good publication ‎

Introduction ‎

‎1-‎ Please highlight the important of folic acid for the human health and the negative effect ‎of its deficiency to spotlight the important of your work ‎

Answer: We add one more highlight about the value of vitamin in human food.

‎2-‎ Please highlight‎ some of the previous reports discussed thermal effect on the retention of ‎folic acid and other micronutrients through the different cooking methods

Answer: We did not find articles where flour preparations were produced for the evaluation of retained folic acid content.

Material and methods

‎1.‎ Samples and reagents

‎1-‎ Lines (88-90): Commercially fortified wheat and corn flours and laboratory fortified ‎wheat and corn flours were evaluated in the laboratory. (what the added value of this ‎sentence please delete it)‎

Answer: A requested correction has been made.

‎2-‎ ‎Lines (91-92): ‎ different brands of wheat flour (WF); different brands of corn flour (CF) ‎‎(please add the name of used brands).‎

Answer: A requested correction has been made.

‎3-‎ Lines (95): ‎item 2.2. (there is no 2.2. please indicate it as 2)‎

Answer: A requested correction has been made.

‎2.‎ Protocol for laboratory folic acid fortification of wheat and corn flour

‎1-‎ Lines (108): ‎a concentration of 1000 μg of folic acid for every 100 g of wheat or corn ‎flour. (Why you exceeded the Brazilian recommended dose; a maximum limit (220 ‎μg.100g-1) for addition of the vitamin in order to avoid exposure of the population to an ‎unnecessary risk (BRAZIL, 2017). ‎

Answer: We had a lot of difficulty with a homogenization step (this was described in the material and methods item) One of the strategies we found was to increase the amount of vitamin added to flour to reduce the effects of this step.

‎2-‎ ‎Lines (109-110): ‎two procedures of homogenization were used (I recommended the first ‎procedure only because it is the simplest and applicable one either for individual or ‎industrial use, so please delete the second procedure).‎

Answer: In this study the two homogenization protocols were applied in order to verify the process difficulties and better condition. We do not find it useful to remove one of these descriptions since these tests are part of the effort made in the study.

‎3-‎ ‎ Lines (124-127): ‎At the end of the fortification procedure, the total time used for the ‎homogenization of the vitamin into the farinaceous matrix, including manual (1 min for ‎each 100 g) and mechanical homogenization (1 min for every 100 g + 2 minutes for 500 ‎g), was of 12 minutes. (Why you mentioned this sentence to compare the spent time and ‎what about the multiple steps and tools is it ok for industrial application??)‎

Answer: We describe in detail the homogenization step due to the absence of folic acid in commercial samples. Our homogenization protocol is experimental and its viability in the industry needs to be evaluated.

‎4-‎ Lines (128): ‎packages to prevent degradation (please describe the full conditions (temp. ‎container material and color, humidity, etc...) you used to prevent degradation ‎ for ‎storage until used).‎

Answer: we replaced the text with: The fortified flours were packaged under the same conditions as those applied in industry.

‎3.‎ Food preparations

‎1-‎ Lines (133): ‎laboratory folic acid fortified wheat flour and corn flour, (which one you ‎used? Homogenized with procedure 1 or 2).‎

Answer: We opted for the first homogenization assay. We clarify this on lines 128 and 309 to 312.

‎4.1. Sample extraction‎

‎1- Lines (200-201): ‎The methodological procedure was a solid-liquid extraction developed ‎and used for plant matrices by Paiva et al. (PAIVA et al., 2012). (Please replace it to; The ‎folic acid was extracted from the samples (fortified wheat flour and corn flour, raw ‎preparations, and cooked preparations) following the methods of PAIVA et al. (2012).‎

Answer: A requested correction has been made.

‎2-‎ Lines (205): ‎ in 5000 g rotation (please replace in with at)‎

Answer: A requested correction has been made.

‎4.2. Chromatographic conditions‎

‎1-‎ Please indicate the full description of HPLC equipment used in this analysis with the ‎country of made, the type of the detector used and wavelength at which the folic acid ‎was detected. Also, indicate the type and specification of the used column.‎

Answer: This description was made on lines 228 through 233.

‎2-‎ Line (218); The column was washed with water and conditioned with acetonitrile ‎‎(Merck-grade HPLC) at the end of the analyses (PAIVA et al., 2012). (Please delete ‎this sentence and move the reference in the right place) ‎

Answer: A requested correction has been made.

‎4.3.Identification and quantification‎

‎1-‎ ‎(1000 μg.mL-1 folic acid) (please replace it with (1000 μg.mL-1 folic acid)‎

Answer: A requested correction has been made.

‎5.True retention ‎

‎(MURPHY; CRINER, GRAY, 1975) (Please correct it to Murphy‎‏ ‏et al‏., 1975)‏

Answer: A requested correction has been made.

Results and discussion

‎1-‎ ‎Line (256-257);‎The methodology applied allowed for there to be retention of folic acid in ‎‎18.2 minutes at maximum absorptivity in 290nm. (Please delete it because you mentioned ‎it in the methodology section).‎

Answer: A requested correction has been made.

‎2-‎ Lines (265-266);‎‏ ‏‎1. Determination of folic acid in commercially fortified and laboratory ‎fortified wheat and corn flours‎‏ ‏‎(please delete it) ‎

Answer: A requested correction has been made.

‎3-‎ ‎‏ ‏Line (268); ‎Fig. 1 Chromatogram o commercially fortified wheat flour and corn flour. ‎‎(Please make it: HPLC chromatogram of commercially fortified wheat flour and corn ‎flour evaluated in July (A) and September (B)).‎

Answer: A requested correction has been made.

‎4-‎ ‎‏ ‏Lines (269-274);‎ ‎ Chromatograms for one of the samples of commercially fortified wheat ‎flour and corn flour evaluated in July (A) and one of the wheat flour and corn flour ‎samples evaluated in September (B) Chromatographic conditions: C18 (150 x 4.6 mm ‎‎5μm) chromatographic column (Allcrom Phenomenex) with the oven at 40 °C, isocratic ‎elution using potassium phosphate buffer (pH 2.0) and methanol (85:15) as mobile phase ‎at a flow of 0.5 mL/min-1 and UV detection with wavelength of 290 nm. (Please delete it ‎and mentioned it in the methodology section‎‏ ‏as general condition for all HPLC analysis‏), ‏and apply it for the other figures‏.‏

Answer: A requested correction has been made.

‎5-‎ Lines (276);‎ (1.000μg.100g-1) ( Please correct it to ‎1000μg.100g-1‎‏) ‏and check it through ‎the paper‏ ‏

Answer: A requested correction has been made.

‎6-‎ ‎ the presentation of the results is not acceptable because of:‎

‎1- Using different units for folic acid in different tables and figures

Answer: All material has been reviewed and as standard units of measurement.

‎2- Content of folic acid in the raw preparations in table 1 is different than in table 3 for ‎example for Bread Loaf is 431 ± 0,4(μg.100g-1) ± SD in table 1 and is 0,43 ± 0,030 (g/100g) ‎‎± DP even 431μg equal 0.000431‬g and what is the meaning of DP. 

‎ Answer: All material has been reviewed and as standard units of measurement. Standard deviation was added.

‎3- Repeated data on the tables and figures, figures 3, 4 derived from table 3, so table 3 is ‎enough

Answer: Chromatogram figures do not allow quantitative measurement. This information is contained only in the tables.

‎4- Table 1 contains extra statistical data have no effect on the study

Answer: We could not identify which statistical information did not belong to the study and which was presented in table 1.

‎5- Cooking conditions columns in Table 3 did not give any information as it as the same ‎mentioned in materials and methods and are not variable studied conditions ‎

Answer: In table 3, we highlight the temperatures for each cooking process already known, the longer or the higher the temperature that decreases the nutrients. Therefore, we consider this information useful for a better interpretation of the data obtained.

‎7-‎ The folic acid peak in figure 2 is not strong peak for trusted quantification of folic acid as ‎the other peaks in the chromatogram are at so near or at the same height. (I highly ‎recommended the preparation of standard curve for folic acid inside each raw ‎preparation. After that, you can use this standard curve for quantification of folic acid ‎inside the raw and cooked preparations.

Answer: It is very common for a color chromatogram to have higher or lower intensity signals for what was not part of the study. The important thing is that the analyte of interest is totally separate from any other substance. In our chromatograms we have the isolated folic acid peak and the other peaks obtained.

 ‎

‎8-‎ Using of standard curve for folic acid inside each raw preparation will verify the retention ‎time for folic acid on the chromatogram and will give you the recovery percent of the ‎solvent used for folic acid extraction from the preparation and it will answer you about ‎the big variation on the folic acid quantities on your different preparations as the ‎extraction method has a strong effect on the recovery content of folic acid. ‎

Answer: In fact the extraction method has significant effect on the success of the analysis. In this sense, we made the best effort to investigate all situations that could make the analysis difficult and we bring this discussion throughout the work.

‎9-‎ The interpretation of discussion needs more supporting results for discussing the ‎variable ‎results ‎against the different conditions of cooked preparation of temperature ‎and ‎foodstuff added per each ‎preparation. ‎ ‎

Answer: we add one more paragraph and more references on the subjects cited.

Reviewer #6:

The MS needs depth revision and English editing. The aims of the study are clear, but the Introduction does not review in depth the problematic and the hypothesis of the MS. I think that the MS is not sufficiently novel and interesting to warrant publication. The work is not original.

Answer: Thank you for your considerations.

Reviewer #7:

The manuscript concerned with determination of Folic acid content in fortified wheat and corn flours and their folic acid

retention in food preparations by using cooking methods: baking, deep

frying and steaming.

In the corn flour preparations, Folic acid retention is high compared to Wheat flour preparations Why?

Answer: We hypothesized for this condition on lines 448 through 455.

The author should do another experiment in which a mixture of folic acid fortified corn and wheat flours are mixed together to testify or investigate the effect of mixing corn with wheat in Foods on the stability or retention of folic acid.

Answer: The description and necessity of this new experiment has not been clarified for us.

Reviewer #8: 

The idea is good.

There is no matching research

The language of the manuscript is good with some simple mistakes.

The methods do not have technical errors that spoil the search.

Answer: Thank you for your considerations.

But: Failure to standardize the expression of concentrations. Makes it difficult to review the results

Data transformation has reduced the variance between treatments, which requires statistical re-analysis

Answer: The necessary corrections have been made regarding the expression of the folic acid concentration obtained

This has led to a change in results but will not affect the overall direction of the findings or final conclusions.

There are some problems with references (in formulation and completeness only).

Answer: A requested correction has been made.

---

## [Decision Letter · Decision Letter 1]

28 Jan 2020

PONE-D-19-26781R1

Folic acid retention evaluation in preparations with wheat flour and corn submitted to different cooking methods by HPLC/DAD.

PLOS ONE

Dear Dra. AZEVEDO,

Thank you for submitting your manuscript to PLOS ONE. After careful consideration, we feel that it has merit but does not fully meet PLOS ONE’s publication criteria as it currently stands. Therefore, we invite you to submit a revised version of the manuscript that addresses the points raised during the review process.

We would appreciate receiving your revised manuscript by Mar 13 2020 11:59PM. To enhance the reproducibility of your results, we recommend that if applicable you deposit your laboratory protocols in protocols.io, where a protocol can be assigned its own identifier (DOI) such that it can be cited independently in the future. For instructions see: http://journals.plos.org/plosone/s/submission-guidelines#loc-laboratory-protocols

We look forward to receiving your revised manuscript.

Kind regards,

Walid Elfalleh, Ph.D

Academic Editor

PLOS ONE

Reviewers' comments:

Reviewer's Responses to Questions

**Comments to the Author**

1. If the authors have adequately addressed your comments raised in a previous round of review and you feel that this manuscript is now acceptable for publication, you may indicate that here to bypass the “Comments to the Author” section, enter your conflict of interest statement in the “Confidential to Editor” section, and submit your "Accept" recommendation.

Reviewer #3: All comments have been addressed

Reviewer #7: All comments have been addressed

Reviewer #8: All comments have been addressed

2. Is the manuscript technically sound, and do the data support the conclusions?

Reviewer #3: Partly

Reviewer #7: Partly

Reviewer #8: Yes

3. Has the statistical analysis been performed appropriately and rigorously? 

Reviewer #3: I Don't Know

Reviewer #7: I Don't Know

Reviewer #8: Yes

4. Have the authors made all data underlying the findings in their manuscript fully available?

Reviewer #3: Yes

Reviewer #7: No

Reviewer #8: Yes

5. Is the manuscript presented in an intelligible fashion and written in standard English?

Reviewer #3: Yes

Reviewer #7: Yes

Reviewer #8: Yes

6. Review Comments to the Author

Reviewer #3: the authors have addressed my comments and explained how they collected their data but the type of column and its specifications used in HPLC analysis weren't mentioned in the chromatographic conditions (3.2) and were deleted in page12 so it has to be added.

Reviewer #7: (No Response)

Reviewer #8: The search is good, its language is intact. Statistical analysis is good. The idea is good.

But there are a few linguistic mistakes. And some references do not match the text or vice versa. Good job in the end.

7. PLOS authors have the option to publish the peer review history of their article (what does this mean?). If published, this will include your full peer review and any attached files.

Reviewer #3: No

Reviewer #7: No

Reviewer #8: Yes: Sabry AbdElMonem AbdAllah

---

## [Author Response · Author response to Decision Letter 1]

29 Feb 2020

Dear editorial below I send as replies of the comments and suggestions of the reviewers. For each comment below follows an answer. We appreciate all the considerations you have made and we seek the ultmost from all that has been requested.

Reviewers' comments:

Reviewer's Responses to Questions

Comments to the Author

1. If the authors have adequately addressed your comments raised in a previous round of review and you feel that this manuscript is now acceptable for publication, you may indicate that here to bypass the “Comments to the Author” section, enter your conflict of interest statement in the “Confidential to Editor” section, and submit your "Accept" recommendation.

Reviewer #3: All comments have been addressed

Reviewer #7: All comments have been addressed

Reviewer #8: All comments have been addressed

Answer: Thank you for your considerations.

2. Is the manuscript technically sound, and do the data support the conclusions?

Reviewer #3: Partly

Reviewer #7: Partly

Reviewer #8: Yes

Answer: Thank you for your considerations.

3. Has the statistical analysis been performed appropriately and rigorously? 

Reviewer #3: I Don't Know

Reviewer #7: I Don't Know

Reviewer #8: Yes

Answer: Thank you for your considerations.

4. Have the authors made all data underlying the findings in their manuscript fully available?

Reviewer #3: Yes

Reviewer #7: No

Reviewer #8: Yes

 Answer: Thank you for your considerations.

5. Is the manuscript presented in an intelligible fashion and written in standard English?

Reviewer #3: Yes

Reviewer #7: Yes

Reviewer #8: Yes

 Answer: Thank you for your considerations.

6. Review Comments to the Author

Reviewer #3: the authors have addressed my comments and explained how they collected their data but the type of column and its specifications used in HPLC analysis weren't mentioned in the chromatographic conditions (3.2) and were deleted in page12 so it has to be added.

Reviewer #7: (No Response)

Reviewer #8: The search is good, its language is intact. Statistical analysis is good. The idea is good.

But there are a few linguistic mistakes. And some references do not match the text or vice versa. Good job in the end.

Answer: References have been revised to meet the journal's standards. Please check the manuscript with markings.

7. PLOS authors have the option to publish the peer review history of their article (what does this mean?). If published, this will include your full peer review and any attached files.

Do you want your identity to be public for this peer review? For information about this choice, including consent withdrawal, please see our Privacy Policy.

Reviewer #3: No

Reviewer #7: No

Reviewer #8: Yes: Sabry AbdElMonem AbdAllah

---

## [Editor Report · Decision Letter 2]

4 Mar 2020

Folic acid retention evaluation in preparations with wheat flour and corn submitted to different cooking methods by HPLC/DAD.

PONE-D-19-26781R2

Dear Dr. AZEVEDO,

We are pleased to inform you that your manuscript has been judged scientifically suitable for publication and will be formally accepted for publication once it complies with all outstanding technical requirements.

With kind regards,

Walid Elfalleh, Ph.D

Academic Editor

PLOS ONE
---

## [Editor Report · Acceptance letter]

27 Mar 2020

PONE-D-19-26781R2 

Folic acid retention evaluation in preparations with wheat flour and corn submitted to different cooking methods by HPLC/DAD. 

Dear Dr. de Paiva Azevedo:

I am pleased to inform you that your manuscript has been deemed suitable for publication in PLOS ONE. Congratulations! Your manuscript is now with our production department. 

With kind regards,

on behalf of

Professor Walid Elfalleh 

Academic Editor

PLOS ONE